# Changes in Demographic Factors' Influence on Regional Productivity Growth: Empirical Evidence from China, 2000–2010

Xiaoxi Wang [1], Yaojun Zhang [1], Danlin Yu [2,*], Xiwei Wu [1] and Ding Li [1]

[1]   School of Sociology and Population Studies, Renmin University of China, Beijing 100872, China; wangxiaoxi2016@ruc.edu.cn (X.W.); zhyaojun@ruc.edu.cn (Y.Z.); wuxiwei@ruc.edu.cn (X.W.); liding@ruc.edu.cn (D.L.)
[2]   Department of Earth and Environmental Studies, Montclair State University, Montclair, NJ 07043, USA
*   Correspondence: yud@mail.montclair.edu

**Abstract:** Improving total factor productivity is an important way for China's economy to avoid the middle income trap. Demographic changes are believed to have significant impacts on productivity growth. Using the census and socioeconomic data of 358 prefecture cities in mainland China, this paper analyzes the changes in the global and local spatial dependence of total factor productivity. We then employ spatial regression methods to investigate the role of changes in population factors in productivity growth in 2000 and 2010. We draw three observations from the analysis. First, population density plays an important role in both years. There is an inverted U-shaped relationship between population density and productivity growth. Second, human capital stock has a significant positive impact while human capital inequality becomes insignificant in 2010. This is likely a result of China's education equality policies. Third, the impact of the aging of workers and their migration status on productivity growth also changed over the decade. Different cohorts of workers and migrants have had different influences on productivity growth because of their different access to higher education. The study provides important insights over how demographic factors impact China's productivity growth.

**Keywords:** demographic factors; total factor productivity; exploratory spatial data analysis (ESDA); spatial regression models

## 1. Introduction

After decades of rapid growth, China's economy has gradually slowed down in recent years. At the same time, inevitable population aging, rising resource and environment tensions, and severe international trading pressure have brought challenges for China's economic development. How to avoid the middle income trap has become a popular topic, and it has attracted increasing attention from both scholars and governmental officials [1,2]. The transition from resource and a labor intensive, low quality model to an advanced, knowledge based development mode, is considered to be the ongoing goal for China's economy in the new century [1,3]. Among various economic factors, total factor productivity is an indicator used to reflect the efficiency and intensity of inputs used in economic production and that have critical impacts on economic fluctuation, growth, and convergence [4–7]. Furthermore, some studies have suggested that total factor productivity growth might even determine how soon a country becomes a high income economy [2,8]. The Chinese government has noticed the importance of improving total factor productivity and regarded it as a major strategy of future economic development. In the *13th Five-Year Plan for Economic and Social Development*, improving total factor productivity has been put on agenda. This was again reemphasized at the *19th CPC National Congress*. It is of great significance to identify the effects of different factors on total factor productivity, which has attracted the attention of many researchers [9–11].

With rapid economic growth, China's population age structure, distribution and quality are also undergoing a series of changes. Specifically, as a result of the decline in total birth rate and the increase in expected life expectancy, China's population is aging quickly. According to the National Bureau of Statistics, the population aged 65 and over in China accounted for more than 7 percent of the total population in 2000, which means that China became an aging society. In 2017, this proportion exceeded 11 percent, and the elderly population exceeded 150 million. In addition, unbalanced regional development leads to massive numbers of young, temporary, migrant workers. In 2015, there were about 247 million temporary migrant workers in China, which accounted for 18% of the total population. A total of 51.1% of these migrants were less than 33 years old [12]. Internal migration also alters the population distribution and age structure in different regions [13]. Furthermore, the improvement in educational level has accumulated a large quantity of highly educated and skilled human capital. In 2016, the total number of students studying in higher education in China reached 36.99 million, and the gross enrollment rate of higher education reached 42.7%. As producers and consumers in economic activities, the population plays an important role in productivity growth. Relevant research has been conducted in the United States [14], Japan [15,16], OECD countries [17] and some other countries [18–20]. The changes in demographic factors in China took place under the guidance of a series of policies and institutions, such as the strict one child policy and unique household registration institution, which have distinct characteristics different from those in other countries. In addition, rapid and dramatic population changes have raised concerns about China's industrial upgrading, technological progress and innovative development [21–23], which are considered as the source of economic productivity growth. However, most of existing studies only focus on one aspect of demographic factors, especially population aging, and comprehensive analyses are relatively limited. The role of changes in demographic factors on economic productivity in China remains a valuable research topic for in depth study.

To this end, the purpose of this paper is to investigate how demographic factors impact on productivity growth changes from 2000 to 2010 in China. We attempt to contribute to the conversation of how demographic influences economic development in China in three aspects. First, largely due to data limitations, previous studies on this topic often focus on the discussion of a single population factor (such as age structure, human capital or migration) [18,20,24] and use observations at a relatively coarse geographic scale (such as provincial or national levels) [3,25–27]. Studies with multiple demographic factors at a finer spatial scale merit further investigation. Given the vast territory and unbalanced development in China, geographical differences within provinces are as vast as across provinces. Data aggregated at large geographic units might mask potentially subtle, yet important, differences that can only be detected at a finer scale. Using census data from 2000 and 2010, we are able to acquire a rich set of population factors over 358 prefecture level cities in China. A prefecture is an administrative unit in China that ranks below province but contains multiple counties. We attempt to employ this data set to help us better understand how population factors impact productivity growth in China. Second, China has established a socialist market economy system since opening up in 1978. Economic development among neighboring cities has become increasingly interconnected. Geographical location becomes an important factor to understand the economic performance of regions due to spatial interaction and interconnection. The spatial effect of data brings a challenge to the traditional data analysis techniques. Spatial data analytical approaches have been proposed in the late 1980s and seen great advancement in recent decades to facilitate the proper analysis of geographically referenced information [28–34]. In the current study, we apply exploratory spatial data analysis (ESDA) to measure the global and local spatial autocorrelation of productivity. We then employ spatial econometrics methods to investigate the role of changes in population factors on productivity growth from 2000 to 2010. Third, with the population changes in China, the relationship between population and economy has undergone major transformations. The traditional demographic dividend [35], which was based on a high proportion of working age labors, is disappearing. Meanwhile, pop-

ulation quality has gained more momentum in driving China's economic development. Comprehensive studies that consider both the quantity and quality of China's population on productivity could contribute to the further understanding of China's contemporary demographic dynamics and how it is entwined with the economic system. Total factor productivity is an important index to measure economic growth equality [36,37]. Therefore, we believe that investigating the impact of demographic factors on total factor productivity from the perspective of a cross temporal comparison will provide better practice for both demographic and economic policies.

The rest of this paper is organized as follows. Section 2 reviews the relative theories and literatures. Section 3 introduces data sources, data preprocessing, variable selection in linear model, and the analysis methods used in the paper. Section 4 presents the findings of the exploratory spatial data analysis on total factor productivity and the spatial regression models. We also discuss our findings extensively in this section. Section 5 concludes with a summary of the findings, presents the limitations of the study, and discusses practical policy implications.

## 2. Literature Review

### 2.1. The Demographic Effects on Productivity

Since Malthus put forward two assumptions of population growth and food production in his famous essay in 1798 [38], how to improve productivity and avoid the Malthusian trap has become a hot research topic. Unified growth theory [8,39,40] explores the theoretical discussion about the relationship between population factors, technological progress, productivity, and economic growth. In the Malthusian epoch, population growth was an engine for accumulation of new ideas, but labor productivity might become stagnant, sometimes even fall back, due to the limited availability of land. When technological innovation passed a threshold point and the growth rate of output is greater than that of the population, the Malthusian trap can then be broken, and economic development transfers to the post-Malthusian phase. In this phase, increased fertility rates and migration from traditional agricultural areas provide sufficient labor forces to encourage a growth in overall productivity. At a certain point, the rising demand for skills and knowledge embodied in human capital leads to the decline in fertility rate, which might temporarily increase the proportion of a working age population (because of the increased number of female workers). With demographic transitions and the accumulation of human capital, the economy will gradually enter the third phase of modern economic growth, which is mainly sustained by technology based productivity growth [8,39]. The impact of demographic factors on productivity growth is multifaceted, complicated, and varying over time, we, hence, contend it will provide more insights if we study this topic from a multifaceted perspective.

Previous studies often look at the influences of population size, age structure, human capital, and migration on productivity. Studies have found the impact of each factor on productivity growth to be complex and, sometimes, even contradictory [16,20,27,41–44]. Kremer [45], in his study on population growth and technological change, found that population size positively influences technological progress over a long historical period. Similar statements can be found in North [46]. Generally, it is easy to understand that a larger population promotes productivity growth due to more potential inventors, more intensive intellectual contacts, greater labor specialization and a bigger size of markets. Some empirical research has argued for the effect of population size on productivity growth as well [16,45]. On the other hand, when population size becomes overly large, a larger population might decrease productivity because of the increased duplication of efforts and decreased available capital stock per capita (the typical crowding effect) [45]. These studies suggest that there is a turning point in the influence curve of population size on productivity growth.

Aging is another important population factor that attracts much attention in the studies on productivity growth. From a physiological perspective, Shephard [41] argues

that, because aging is associated with progressive changes in power, thermoregulation, reaction speed and the acuity of the special senses, the productivity of the elderly will decrease, particularly in self paced activities. In addition, compared to younger people, the aged are believed to be less innovative and receptive to new technologies, which blocks productivity improvement [47]. On the other hand, Gordo and Skirbekk [24] find that older workers are better able to adapt to technological changes. Ang and Madsen [42] point out that aging need not to be a drag on productivity. They argue that experience and knowledge developed tends to improve older workers' productivity.

Although it is widely believed that human capital investment is conducive to long-term economic growth, a consensus is yet to be reached about the impact of human capital on productivity [10,43,44]. Benhabib and Spiegel [43] argue that human capital helps workers to create, learn and use more advanced and newer technologies, thus it is helpful to improve productivity by promoting technology progress and diffusion. Miller and Upadhyay [9] point out that the effect of human capital on productivity in low income countries is changeable. It moves from negative to positive as the country moves from low to high levels of openness. Pritchett [44] even shows a significant negative relationship between human capital and total factor productivity. Gong [3] focuses on both the stock and inequality of human capital and believes that high quality human capital has a positive effect on productivity by affecting technology innovation, and human capital inequality has a negative effect on productivity by affecting technology diffusion.

The dualistic economic structure theory proposed by Arthur Lewis [48] is often used to explain the phenomenon of migration on productivity. It is assumed that urban sectors have higher productivity. Rural–urban migration could promote overall economic productivity. In other words, population migration is essentially a reallocation of human resources, and it brings obvious impacts on productivity for both inflow and outflow areas, which is a Pareto improvement for individuals, enterprises, and countries [12,49]. However, the empirical research on this topic has failed to reach a consensus [20,27]. Paserman [19] argues that, because of the different characteristics of migrants, especially their skill levels, they play various roles in the dissemination of new ideas and technologies. Low skilled migrants have little impact on technological diffusion in the receiving areas, and an excessive dependence on manual labor is not conducive for enterprises to improve production efficiency. In summary, the influence mechanism of migration on productivity is mainly in technology adoption, human capital formation, innovation process and knowledge spillovers [19,50,51].

### 2.2. Productivity Measure and Other Control Variables

There are many indices for measuring overall economic productivity, such as per capita output, labor productivity (LP), multifactor productivity (MFP) and total factor productivity (TFP). Among them, we contend that total factor productivity might provide a more comprehensive understanding of the productivity changes caused by the different factors input of production. Total factor productivity is a preferable measure to productivity in many previous studies [3,6,19,52]. To fully understand how population factors impact on productivity, we need to control for other factors affecting productivity. Studies suggest that openness, trade orientation, foreign direct investment, industry structure, government expenditure, and geographical location are important factors [3,9–11,52–54].

### 2.3. The Advance and Application of Spatial Data Analysis Methods

In empirical studies of the social sciences, regional socioeconomic data are often used in many studies [3,6,27,52]. When conventional econometric methods are used, these studies implicitly assume that the observations collected with geographical information are independent and there is no connection between regions. However, according to the First Law of Geography [55], spatial data are inherently interconnected and interdependent between different spatial units, and the strength of such interdependence depends on geographical distance. This suggests that socioeconomic development in a certain region might

very likely be closely related to the surrounding regions. Such interconnectedness and interdependence are often collectively called spatial autocorrelation among observations collected over geographic space [28,56]. The existence of spatial autocorrelation violates the independence assumption of observations in traditional statistical analysis. For instance, when investigating relationships between demographic factors and productivity, if we employ the common OLS regression estimator for geographic data, we might produce potentially unreliable and even misleading results due to the potential existence of spatial autocorrelation in the regression residuals. Instead, the estimator based on the maximum likelihood (ML) method is usually proposed as an effective alternative [28]. Some existing studies have noticed the limitations of using traditional regression methods to analyze spatial data [6,49] and scholars have tried to use spatial econometric methods to explain the influence mechanism on productivity [52,54].

There are two objectives that we intend to achieve by employing spatial data analysis methods in the investigation of demographic factors' impacts on regional productivity. First, we attempt to investigate the interconnectedness of productivity in different regions to tell a more informative story about the development process of productivity. Second, the application of spatial regression methods in empirical research helps us capture and control spatial effects, hence providing a more convincing result. With more reasonable regression results, we might have a better understanding of the influence mechanism of population factors on productivity growth at a city level.

### 3. Data and Methodology

*3.1. Data Sources and Preprocess*

Since the first population census in 1953, China has conducted seven population censuses. Although the most recent census was conducted in 2020, data needed for calculating variables such as the number of net immigrants and human capital Gini coefficient, for the current analysis, have not yet been released at the city level. For this reason, the census data in 2000 and 2010 are used instead. Admittedly, population dynamics and economic productivity in China have experienced great changes over the decade from 2010 to 2020; we hope the current study still sheds light on how demographic factors impact productivity in China during its fastest growing periods of both population and economic dynamics. In addition, it is often difficult (if not impossible) to obtain detailed population data in non-census years, particularly at city level. Our study attempts to focus on census data because censuses offer significant demographic information and provide a great opportunity to study topics concerning population factors at finer spatial scales.

The original data used in the paper are compiled from the following sources. For demographic information, we include total resident population, total registered population, population aged 50–59, population with different education levels, and total working population. For other socioeconomic control factors, and data to calculate total factor productivity, we include investment in fixed asset, the proportion of secondary and tertiary industries, government expenditure, foreign direct investment, and land area of a city. The data on population factors are mainly from *Tabulation on the Population Census of China by County* in 2000 and 2010. These data sets are published by the China Statistics Press. It collects detailed information on population statistics at both the county and city levels, such as the total resident population, total household registration population, population by sex, population by residence, population by age group, population by educational level, population by marriage status and other indexes such as family size and employment. Second, economic indicators, such as city level GDP, total investment in fixed assets, industrial structure, and government expenditure, come from the *China City Statistical Yearbook* and *China Statistical Yearbook for Regional Economy*. The yearbooks of some western provinces also offer a useful supplement in cases when data is missing in the above two yearbooks. Third, due to the serious lack of foreign direct investment data in some cities, we use an alternative indicator by summing up the foreign capital of industrial enterprises in each city. The original data are collected from the *China Industry Business Performance Data*,

which is a large data source that includes the capital composition of industrial enterprises. All price related economic data are converted to comparable prices based on the price level in 2000.

### 3.2. TFP and Demographic Variables

In the following analysis, most variables can be extracted directly from the collected data. Some variables need to be calculated, including the total factor productivity, human capital Gini coefficient and the number of net immigrants. Specifically, the measurement of total factor productivity is based on conventional Solow residual value method with no scale return assumption. This is a preferable method and has been employed in many studies [3,20,42,57]. The idea of Solow residual value method is relatively straightforward for understanding total factor productivity. We first estimate the total production function. We then eliminate the contribution of factor input in economic output. What is left is the estimation of total factor productivity, indicating the contribution of technological progress, resource reallocation, institutional improvement, and other factors to economic output. It is often assumed that the total production function takes the Cobb–Douglas form:

$$Y = AK^{\alpha}L^{\beta} \tag{1}$$

where $Y$ denotes the total output (GDP in our case), $K$ denotes the capital stock and is calculated by applying the perpetual inventory method with 5% annual depreciation [49,57], $L$ denotes labor and $A$ denotes total factor productivity. Take the natural logarithm on both sides and we get:

$$\ln Y = \ln A + \alpha \ln K + \beta \ln L \tag{2}$$

With relevant data and an ordinary least square regression, we can estimate $\alpha$ and $\beta$. We assume that the return on scale remains the same, which means that the coefficients for $K$ and $L$ shall sum to 1. To calculate TFP, we then transform the estimated coefficients, $\alpha$ and $\beta$, to be:

$$\alpha^* = \alpha/(\alpha + \beta), \ \beta^* = \beta/(\alpha + \beta) \tag{3}$$

Then total factor productivity is measured using formula:

$$A = Y/(K^{\alpha^*}L^{\beta^*}) \tag{4}$$

To evaluate the demographic factors' impact on *TFP*, we consider both the quality and quantity of the population, as well as the mobility of the population. The quality of population is represented by the number of people with college and above level of education, and the portion of population that is 50–59 years old (the older working population in China). Total number of residents is used to represent the quantity of population. Mobility of the population is represented by the number of net immigrants for each prefecture. The number of net immigrants is calculated by subtracting the number of household registered population from the total resident population. The unique household registration system in China requires each person to register his/her household address, which is usually their birthplace, so the number of net immigrants in cities can be roughly obtained by comparing the difference between registered population and current residents.

The descriptive statistics for the key variables in 2000 and 2010 are reported in Table 1. From 2000 to 2010, the mean of total factor productivity (*TFP*) has almost doubled from 46.443 to 83.642, indicating that China has promoted overall economic productivity significantly over this period. When it comes to population factors, the total resident population (*POP*) increases slightly due to strict one child policy and declining fertility rate. The portion of elderly population is increasing rapidly. The mean of proportion of population aged 50–59 (*OLD*) in 2010 is 11.624, with an increase by nearly 3 percentage points from 2000. Human capital stock, represented by the number of people with college education and over (*HCS*), increases remarkably. Migration between regions becomes more frequent as well,

which is represented by the changes in minimum and maximum value of the number of net immigrants (*MIG*) from 2000 to 2010.

**Table 1.** The descriptive statistics for the key variables.

| Variable | Unit | Mean | Std.Dv | Min | Max |
|---|---|---|---|---|---|
| *TFP2000* | | 46.443 | 18.413 | 8.851 | 134.018 |
| *POP2000* | million | 3.466 | 2.868 | 0.077 | 30.513 |
| *OLD2000* | % | 8.509 | 1.431 | 2.371 | 12.718 |
| *HCS2000* | thousand | 122.748 | 209.759 | 1.007 | 2284.850 |
| *MIG2000* | million | 0.023 | 0.572 | −1.208 | 5.786 |
| *TFP2010* | | 83.642 | 37.226 | 22.519 | 256.869 |
| *POP2010* | million | 3.720 | 3.197 | 0.076 | 28.846 |
| *OLD2010* | % | 11.624 | 2.581 | 4.368 | 19.362 |
| *HCS2010* | thousand | 328.919 | 562.886 | 6.273 | 6177.772 |
| *MIG2010* | million | −0.013 | 1.107 | −4.304 | 8.834 |

*3.3. Control Variables and Models*

Other than the demographic factors, there are also socioeconomic factors that likely influence on total factor productivity. To fully understand how demographic factors influence *TFP*, we also need to control relevant socioeconomic factors that could influence *TFP*. Literature review suggests that, in China, governmental investment (represented by fixed assets investment), industrial structures (portion of secondary and tertiary industries in the economy), government financial expenses, and the overall openness of a prefecture are commonly believed to have significant impact on total factor productivity [10,11,52,58–63]. Details of the control variables follow.

Fixed assets investment (*INV*): fixed assets investment is an important link in the process of economic reproduction. Reasonable investments are conducive to the improvement in per capita capital stock and encourage technological progress.

Industrial structure (*SEC* and *TER*): economic development is a process of gradual transformation of industrial structure. Industrial agglomeration can promote division of labor, technology exchange and scale effect. Industrial diversification and density can promote the production efficiency of industries to increase productivity. We use the proportion of the secondary and tertiary industry to examine the impact of the industrial structure on productivity growth.

Government expenditure (*GOV*): local government has played an important role in promoting regional economic development in China. Every year, the government invests heavily in infrastructure construction, public safety, education and scientific research and social security. This helps speed up the flow of production factors, promotes the diffusion of knowledge and technology, reduces market transaction costs, and improves the efficiency of resource allocation.

Openness (*FDI*): with expanding openness, foreign capital has participated in China's economic activities through direct investment or other forms of investment. Advanced production technology and management experience have been introduced, and domestic enterprises have greatly improved production efficiency through "learning by doing".

Based on our preliminary exploratory analyses, the relationships between TFP and the demographic factors, as well as other socioeconomic influencing factors, are a combination of approximately exponential and linear. The influencing factor model of total factor productivity is then constructed as follows:

$$\ln(TFP) = \beta_0 + \beta_1 POP + \beta_2 OLD + \beta_3 \ln(HCS) + \beta_3 MIG + \beta_4 \ln(INV) + \beta_5 SEC + \beta_6 TER + \beta_7 \ln(GOV) + \beta_8 \ln(FDI) + \varepsilon \quad (5)$$

In the above formula, the outcome variable ln(*TFP*) is the natural logarithmic value of total factor productivity. The growth rate of total factor productivity is obtained through

this transformation because of a likely exponential relationship between impact factors and total factor productivity [3,9,20].

In addition, as the size of population increases, the marginal benefit of added labor force could potentially be countered by limited economic carrying capacity. We intend to examine whether there is an inverted U-shaped relationship between population size and productivity growth. Moreover, as argued in Klasen and Nestmann [26] and Rizov and Zhang [64], when evaluating population's impact on productivity, population density, rather than size, could be the critical influencing factor on productivity growth. We intend to replace population size with density to see the difference. Although education quality is proxied in the model with the number of people who have college and above level education, it is also noted that education levels in China are vastly unequal across different regions. Other than examining the education level's impact on total factor productivity, we also introduce the human capital Gini coefficient (*HCI*) to represent human capital inequality and model its impact on total factor productivity. According to Thomas, Wang and Fan [65] and Gong [3], human capital Gini coefficient is formulated as follows:

$$HCI = \frac{1}{H} \sum_{i=2}^{n} \sum_{i=1, \, j=1}^{n} |E_i - E_j| P_i P_j \qquad (6)$$

where $E_i$ and $E_j$ denote the years of different education levels with education level at illiteracy $E_1 = 0$, primary $E_2 = 6$, junior $E_3 = 9$, senior and secondary technical $E_4 = 12$, and junior college and above $E_5 = 16$, based on China's current educational system (the years of schooling at each level of education). $P_i$ and $P_j$ denote the proportion of the population at each education level; and H denotes the weighted education years and takes the form:

$$H = E_1 P_1 + E_2 P_2 + E_3 P_3 + E_4 P_4 + E_5 P_5 \qquad (7)$$

The *HCI*, just as the Gini coefficient is often used in gauging economic inequality, measures the education distribution among a prefecture's population, the coefficient ranges from 0 to 1 with higher coefficient suggesting higher inequality [65]. In his work, Gong [3] suggests that higher human capital inequality will negatively affect technology diffusion, hence, *TFP*, which is different from human capital quality's influence, which is usually considered a positive factor for *TFP* because it promotes technological introduction and innovation.

To model all these effects, we call the model in Equation (5) the base model (model 1). From the base model, model 2 adds the squared term of population size (*POP_sq*). In model 3, we replace *POP* and *POP_sq* in model 2 with population density (*PD*) and its squared term (*PD_sq*). At last, model 4 adds the human capital Gini coefficient (*HCI*).

### 3.4. Methodology
3.4.1. Exploratory Spatial Data Analysis (ESDA)

Exploratory spatial data analysis is an effective means to explore the potential spatial effects and patterns in geographic data [66]. We use the global and local version of the Moran's Index to explore the spatial patterns of total factor productivity at the prefecture level in China. The Moran's Index takes the form [66]:

$$I = \frac{n \sum_{i=1}^{n} \sum_{j \neq i}^{n} w_{ij} (x_i - \overline{x})(x_j - \overline{x})}{\sum_{i=1}^{n} (x_i - \overline{x})^2 \sum_{i=1}^{n} \sum_{j \neq i}^{n} w_{ij}} \qquad (8)$$

where $x$ represents the observed values and $w_{ij}$ refers to the spatial weight between two spatial units $i$ and $j$. The spatial weight matrix $W_n$ is an $n * n$ matrix that defines the spatial influence strength between spatial units and their neighbors. The diagonal elements of the matrix are set to zero and they are usually row standardized to ensure all the weights are between 0 and 1. In this paper, we construct two types of spatial weight matrices: simple binary spatial matrix ($W_{adj}$) with Queen contiguity rule and inverse distance weight

spatial contiguity matrix ($W_{dis}$), to test how different spatial configurations impact on spatial effects.

To assess the regional structure of spatial autocorrelation, we turn our attention to the local version of Moran's Index and the Moran's scatterplot. Moran scatterplot plots the spatial lag values against the original values. The scatterplot is divided into four different quadrants according to the four types of local spatial association. HH means a region with high value surrounded by regions with high values and HL means a region with high value surrounded by regions with low values [66]. Although the Moran scatterplot can be used to detect atypical regions, it does not give any indications of significant spatial clustering or hot spots. Local Moran's I [67] is produced to visualize potential "clusters" or "hot spots". It takes the form:

$$I_i = \frac{(n-1)(x_i - \overline{x})}{\sum_{j \neq i}^{n} (x_j - \overline{x})} \sum_{j \neq i}^{n} w_{ij}(x_j - \overline{x}) \qquad (9)$$

### 3.4.2. Spatial Regression Models

When spatial data is involved in a regression analysis, the traditional estimator, ordinary least squares, is no longer valid because of possible spatial autocorrelation in regression residuals. Anselin [28–30], among many others, proposed the maximum likelihood estimator as an alternative, since it is not restricted by independent regression residuals. Based on the possible sources of spatial autocorrelation in regression residuals, there are two types of spatial autoregressive models that are often considered in empirical studies [60]. If the source is from a spatially autocorrelated dependent variable, we have a spatial lag specification. If, however, spatial autocorrelation in the residual is from an omitted but spatially autocorrelated explanatory variable, then we will have a spatial error specification. Elhorst [68] has outlined a taxonomy of different spatial models. We focus on the spatial lag (SLM) and spatial error (SEM) specification in our current study. The spatial lag autoregressive model can be specified as follows:

$$y = \rho W y + \beta X + \varepsilon \qquad (10)$$

where $y$ is the dependent variable, $W$ is a spatial weight matrix describing the spatial influence among different areas, $Wy$ is a spatially lagged dependent variable, and $\rho$ is the coefficient of $Wy$ measuring the spatial dependence on dependent variable. $X$ is a matrix of independent variables and a constant term, $\beta$ is the coefficient vector of independent variables, and $\varepsilon$ is a well behaved error term with mean zero. The spatial error autoregressive model has the following expression:

$$\begin{aligned} y &= \beta X + \varepsilon \\ \varepsilon &= \lambda W \varepsilon + \mu \end{aligned} \qquad (11)$$

where $\lambda$ is the coefficient of the error measuring the spatial mismatch, and $\mu$ is a well behaved error term. The other symbols are defined as in the spatial lag model. As argued in Anselin [28] and Anselin, Bera, Florax and Yoon [29], Lagrange multiplier (LM) robust diagnostic tests based on ordinary least square (OLS) residuals can help us choose a more appropriate model between SLM and SEM. The model that has a larger and significant robust LM statistics often is the more appropriate specification. Since the spatial autoregressive model is estimated via the maximum likelihood estimator, the likelihood based information criteria, such as the Akaike information criterion, is used for the test of goodness of fit and comparison among the models. The model with smaller AIC is considered the better fit for the data [60]. All the calculation of the above statistics and models is conducted using the SPDEP package in R [31,32], freely available through the R project website.

## 4. Results and Discussion

### 4.1. Examining Spatial Autocorrelation with ESDA

Since the reform and opening up in 1978, economic development has become the central task of the Chinese government. During the reform period, the exchange of production factors—including labor, capital, resources, and technologies—between regions became increasingly frequent. This has led to a transformation of the regional economy, from self-governing to interdependence. We calculated global and local Moran's I to investigate the change in the spatial dependence of total factor productivity from 2000 to 2010, in an attempt to capture such a change. The results of global Moran's I and the responding significance tests with different spatial weight matrices are reported in Table 2.

**Table 2.** Significance test of global Moran's I for total factor productivity.

|  | $(W_{adj})$ | | $(W_{dis})$ | |
| --- | --- | --- | --- | --- |
|  | **I** | ***p*-Value** | **I** | ***p*-Value** |
| TFP2000 | 0.405 | $<2.2 \times 10^{-16}$ | 0.452 | $<2.2 \times 10^{-16}$ |
| TFP2010 | 0.452 | $<2.2 \times 10^{-16}$ | 0.453 | $<2.2 \times 10^{-16}$ |

It can be seen that the Moran's *I* values of total factor productivity are significantly positive for both years, indicating there is a strong similarity or dependence between the total factor productivity of neighboring cities in these two years in China. The difference between using a different weight matrix, however, remains negligible, which suggests the spatial autocorrelation of productivity in China is the result of increased interconnectedness among cities after the economic reform, regardless of how a neighborhood is defined. When looking at the change from 2000 to 2010, we find that the Moran's I of total factor productivity becomes larger, which means a general trend of increasing spatial autocorrelation in these ten years (increased interconnectedness). The other interesting finding is that the difference in Moran's I in 2010 with different weighting strategies is less than that in 2000. In other words, the impact of geographical distance on the spatial dependence of productivity becomes smaller in 2010. This might be explained by the improvement in China's transport facilities leading to the decrease in the impediment effect of geographical distance, which is a relatively important factor in the exchange of technologies, talents and production factors. In 2000, the road mileage in China was 1.4 million km and railway mileage 58,700 km. These two figures increased to 4 million km and 90,000 km in 2010. Moreover, the high speed railway construction project was launched in 2008, and its mileage reached beyond 5000 km in 2010, which promoted a more integrated regional productivity landscape.

The local Moran's I is mapped in Figures 1 and 2. From the figures, we can see that the number of L-L clustered cities in 2010 is less than that in 2000, which is opposite to the H-H type. The rapid growth in total factor productivity in Xinjiang has led to the change from a gradient pattern decreasing from east to west in 2000 to a sandwich like pattern in 2010. The grand western development program initiated in 2000 helped Xinjiang develop at a high speed. In addition, Xinjiang is a province with net immigrants in western China and the number of net immigrants was more than 1.5 million in 2010. Xinjiang also has 10,635 people with junior college and above degrees per 100,000 persons in 2010, ranking sixth among the 31 provincial administrative units in mainland China. Population immigration and the large proportion of people with higher education might be a critical factor for Xinjiang to promote its productivity. We will examine the mechanisms for productivity growth in the following section.

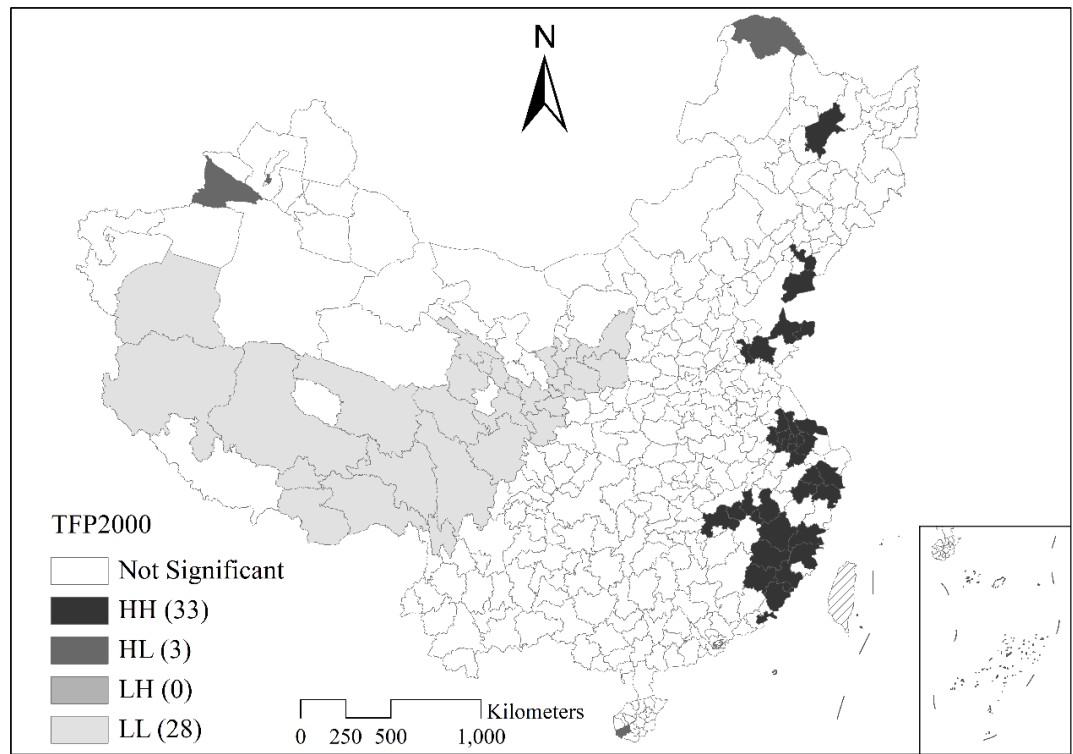

**Figure 1.** Local Moran's I significance maps of total factor productivity in 2000.

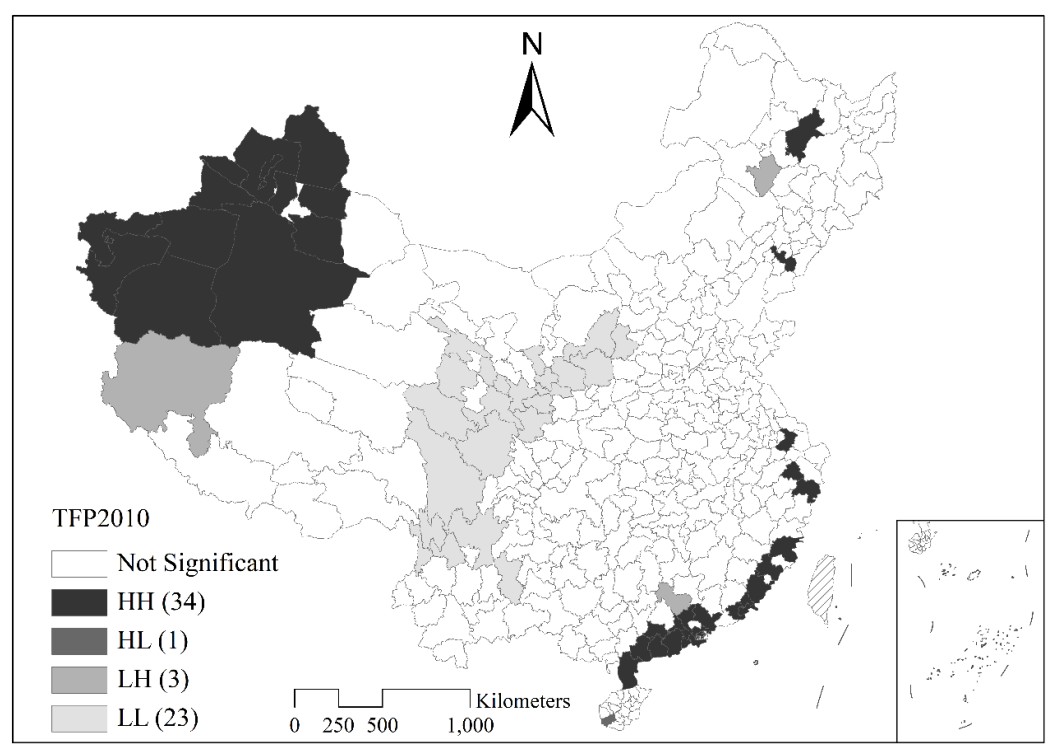

**Figure 2.** Local Moran's I significance maps of total factor productivity in 2010.

*4.2. Regression Results*

For the regression analysis, we conducted a regular OLS estimation with the four demographic factors, namely, population size (POP), aging of workers (OLD), human capital stock (HCS) and migration (MIG), and other control variables, and report the results

for 2000 and 2010 in Table 3. The Lagrange multiplier (LM) tests were then conducted with different weight matrices and reported in Table 3.

**Table 3.** OLS results for Model 1 in 2000 and 2010.

| | **2000** | | | **2010** | | |
| | Coefficient | Std. Error | *t* Value | Estimate | Std. Error | *t* Value |
|---|---|---|---|---|---|---|
| (Intercept) | 4.703 *** | 0.351 | 13.411 | 8.585 *** | 0.502 | 17.086 |
| POP | −0.002 | 0.009 | −0.188 | 0.005 | 0.01 | 0.533 |
| OLD | −0.003 | 0.013 | −0.226 | 0.020 ** | 0.008 | 2.534 |
| lnHCS | 0.192 *** | 0.034 | 5.726 | 0.462 *** | 0.048 | 9.628 |
| MIG | 0.014 | 0.036 | 0.401 | 0.072 *** | 0.019 | 3.765 |
| lnINV | −0.182 *** | 0.031 | −5.823 | −0.301 *** | 0.049 | −6.118 |
| SEC | 0.008 *** | 0.002 | 4.547 | −0.003 | 0.002 | −1.531 |
| TER | −0.004 | 0.003 | −1.648 | −0.016 *** | 0.003 | −5.139 |
| lnGOV | 0.013 | 0.025 | 0.516 | −0.160 *** | 0.054 | −2.931 |
| lnFDI | 0.031 *** | 0.004 | 7.055 | 0.042 *** | 0.005 | 7.791 |
| Model summary | | | | | | |
| Adjusted R-squared: | | 0.408 | | | 0.399 | |
| F-statistic: | | 28.32 on 9 and 348 df, *p* = 0.000 | | | 27.37 on 9 and 348 df, *p* = 0.000 | |
| Spatial autocorrelation test for residuals | | | | | | |
| | $W_{adj}$ | $W_{dis}$ | | $W_{adj}$ | $W_{dis}$ | |
| Moran's I | 0.258 *** | 0.287 *** | | 0.328 *** | 0.326 *** | |

Notes: the dependent variables are lnTFP2000 in 2000 and lnTFP2010 in 2010; *** and ** denote 1% and 5% levels of significance, respectively.

Results from Table 4 reveal that there is significant spatial autocorrelation in the OLS's regression residuals. As the LM tests suggest, the likely cause for the spatial autocorrelation in the regression residuals is from the spatially autocorrelated dependent variable. According to selection criteria proposed in Anselin [28] and Elhorst [68], the spatial lag autoregressive model might be a more appropriate alternative, indicating a strong spatial dependence and connection among neighboring cities in terms of productivity. We then present the spatial lag estimation results in Table 5. The additional models' results are presented in Table 6.

**Table 4.** LM tests for OLS models with different spatial weight matrices.

| | **2000** | | **2010** | |
| | $W_{adj}$ | $W_{dis}$ | $W_{adj}$ | $W_{dis}$ |
|---|---|---|---|---|
| LM-err | 59.387 *** | 56.27 *** | 96.017 *** | 72.518 *** |
| LM-lag | 64.834 *** | 61.084 *** | 118.82 *** | 93.153 *** |
| RLM-err | 3.384 * | 2.704 | 2.209 | 0.326 |
| RLM-lag | 8.831 *** | 7.517 *** | 25.012 *** | 20.961 *** |

Note: *** and * denote 1% and 10% levels of significance, respectively.

By observing the results from Tables 5 and 6 and comparing them with Table 3, we have a few interesting results. First, as shown by the log likelihood and AIC values in Table 5, the spatial regression models have significantly improved over the OLS models, regardless of which spatial weight matrix we use. The log likelihood values increase by nearly 30 (40) in 2000 (2010) and AIC values decrease by nearly 50 (70) in 2010. The ρ values in spatial lag models are significant at 99% confidence level, indicating that a strong spatial autocorrelation does exist and the total factor productivity in the surrounding regions has a strong positive diffusive effect. The spatial autocorrelation tests for the spatial lag model's residuals suggest the residuals are no longer autocorrelated, which means that the spatial effect is well dealt with by adding the spatial lag terms. In addition, compared with the results of OLS models (Table 3), the coefficients of the independent variables have changed in different directions in the spatial lag models. For example, the coefficient of

population size (POP) turns to positive in the spatial lag model for 2000 and the coefficient of migration (MIG) becomes smaller in the spatial lag models for both years. The correction of the coefficients by the spatial regression models provides potentially a more realistic understanding of the impact of each explanatory variable.

**Table 5.** Spatial regression results for Model 1 in 2000 and 2010.

| | 2000 | | 2010 | |
|---|---|---|---|---|
| | $W_{adj}$ | $W_{dis}$ | $W_{adj}$ | $W_{dis}$ |
| (Intercept) | 3.263 *** | 3.343 *** | 5.402 *** | 5.755 *** |
| | (0.381) | (0.372) | (0.523) | (0.522) |
| POP | 0.005 | 0.004 | 0.003 | 0.002 |
| | (0.008) | (0.008) | (0.008) | (0.008) |
| OLD | −0.011 | −0.013 *** | 0.016 ** | 0.014 ** |
| | (0.012) | (0.012) | (0.007) | (0.007) |
| lnHCS | 0.172 *** | 0.166 *** | 0.387 *** | 0.388 *** |
| | (0.031) | (0.031) | (0.042) | (0.042) |
| MIG | 0.007 | 0.008 | 0.037 ** | 0.042 ** |
| | (0.032) | (0.032) | (0.016) | (0.017) |
| lnINV | −0.194 *** | −0.184 *** | −0.302 *** | −0.298 *** |
| | (0.028) | (0.028) | (0.042) | (0.043) |
| SEC | 0.007 *** | 0.008 *** | −0.001 | −0.002 |
| | (0.002) | (0.002) | (0.002) | (0.002) |
| TER | −0.003 | −0.003 | −0.012 *** | −0.012 *** |
| | (0.002) | (0.002) | (0.003) | (0.003) |
| lnGOV | 0.025 | 0.020 | −0.050 | −0.064 |
| | (0.022) | (0.022) | (0.047) | (0.048) |
| lnFDI | 0.019 *** | 0.019 *** | 0.032 *** | 0.033 *** |
| | (0.004) | (0.004) | (0.005) | (0.005) |
| ρ | 0.439 *** | 0.412 *** | 0.476 *** | 0.428 *** |
| | (0.054) | (0.051) | (0.051) | (0.050) |
| Model summary | | | | |
| Log likelihood: | −43.903 | −43.422 | −62.538 | −68.450 |
| Log likelihood for OLS: | −72.007 | −72.007 | −105.840 | −105.840 |
| AIC: | 111.81 | 110.84 | 149.08 | 160.90 |
| AIC for OLS: | 166.01 | 166.01 | 233.68 | 233.68 |
| Spatial autocorrelation test for residuals | | | | |
| Moran's I | −0.009 | −0.002 | −0.025 | −0.050 |
| *p*-value | 0.579 | 0.454 | 0.751 | 0.892 |

Notes: the dependent variables are lnTFP2000 in 2000 and lnTFP2010 in 2010; figures in the parentheses indicate standard errors; *** and ** denote 1%, and 5% levels of significance, respectively.

**Table 6.** Spatial regression results for Model 2–4 in 2000 and 2010.

| | Model 2 | | Model 3 | | Model 4 | |
|---|---|---|---|---|---|---|
| | 2000 | 2010 | 2000 | 2010 | 2000 | 2010 |
| (Intercept) | 3.339 *** | 5.757 *** | 3.449 *** | 5.712 *** | 3.463 *** | 5.325 *** |
| | (0.37) | (0.524) | (0.368) | (0.506) | (0.372) | (0.482) |
| POP (PD) | 0.017 | 0.002 | 0.323 *** | 0.193 * | 0.329 *** | 0.207 ** |
| | (0.015) | (0.017) | (0.116) | (0.099) | (0.115) | (0.098) |
| POP_sq (PD_sq) | −0.001 | 0.000 | −0.242 *** | −0.095 ** | −0.247 *** | −0.111 ** |
| | (0.001) | (0.001) | (0.079) | (0.046) | (0.079) | (0.045) |
| OLD | −0.013 | 0.014 ** | −0.019 | 0.014 * | −0.018 | −0.001 |
| | (0.012) | (0.007) | (0.012) | (0.007) | (0.012) | (0.008) |
| lnHCS | 0.152 *** | 0.387 *** | 0.166 *** | 0.378 *** | 0.136 *** | 0.313 *** |
| | (0.033) | (0.046) | (0.028) | (0.04) | (0.032) | (0.033) |
| MIG | 0.007 | 0.042 ** | 0.076 * | 0.059 *** | 0.070 | 0.069 *** |
| | (0.032) | (0.017) | (0.043) | (0.022) | (0.043) | (0.021) |
| HCI | | | | | −0.455 ** | 0.106 |
| | | | | | (0.23) | (0.38) |

**Table 6.** *Cont.*

| | Model 2 | | Model 3 | | Model 4 | |
|---|---|---|---|---|---|---|
| | **2000** | **2010** | **2000** | **2010** | **2000** | **2010** |
| lnINV | −0.186 *** | −0.298 *** | −0.188 *** | −0.311 *** | −0.180 *** | −0.285 *** |
| | (0.028) | (0.043) | (0.027) | (0.043) | (0.028) | (0.042) |
| SEC | 0.008 *** | −0.002 | 0.007 *** | −0.002 | 0.007 *** | 0.000 |
| | (0.002) | (0.002) | (0.001) | (0.002) | (0.001) | (0.002) |
| TER | −0.002 | −0.012 *** | −0.003 | −0.012 *** | −0.003 | −0.008 *** |
| | (0.002) | (0.003) | (0.002) | (0.003) | (0.002) | (0.003) |
| lnGOV | 0.022 | −0.064 | 0.022 | −0.043 | 0.040 * | −0.003 |
| | (0.022) | (0.048) | (0.021) | (0.047) | (0.023) | (0.045) |
| lnFDI | 0.019 *** | 0.033 *** | 0.018 *** | 0.032 *** | 0.017 *** | 0.026 *** |
| | (0.004) | (0.005) | (0.004) | (0.005) | (0.004) | (0.005) |
| ρ | 0.410 *** | 0.428 *** | 0.409 *** | 0.429 *** | 0.384 *** | 0.450 *** |
| | (0.051) | (0.05) | (0.051) | (0.05) | (0.052) | (0.05) |
| Model summary | | | | | | |
| Log likelihood: | −42.857 | −68.45 | −38.881 | −66.22 | −36.99 | −64.417 |
| Log likelihood for OLS: | −71.212 | −105.579 | −66.724 | −102.569 | −60.359 | −104.102 |
| AIC: | 111.71 | 162.9 | 103.76 | 158.44 | 101.98 | 156.83 |
| AIC for OLS: | 166.42 | 235.16 | 157.45 | 229.14 | 146.72 | 234.2 |

Notes: the dependent variables in all models are lnTFP2000 in 2000 and lnTFP2010 in 2010; figures in the parentheses indicate standard errors; ***, ** and * denote 1%, 5% and 10% levels of significance, respectively.

Second, population size seems to have no effect on productivity growth in both years. The population size in China is large. The average population size of cities in 2010 is 3.72 million. The least populous city is Shennongjia in Hubei Province, which had more than 70 thousand people in 2010. The reason why population size does not improve productivity might be that, in the post-Malthusian period, innovation and technological progress mainly come from the secondary and tertiary industries rather than agriculture, which do not depend on population size. Another explanation, from Klasen and Nestmann [26], is that population density, instead of size, plays an important role in knowledge creation and diffusion, market expansion and technological progress. We will examine this argument in further exploration. On the contrary, human capital stock (HCS) is found to have a significant positive impact on productivity growth in both years. The coefficient in 2010 is larger than that in 2000, indicating an increasing growth effect of human capital stock on productivity. The rapid accumulation of human capital might be largely due to the expansion of China's higher education. According to official statistics, the annual enrollment in higher education in China has increased from 3.90 million in 2000 to 9.56 million in 2010, an increase of 2.5 times within a decade. The other two population factors, aging and migration, have changed their impact on productivity growth with varying symbols and significance levels. As shown in Table 5, the coefficient of aging workers (OLD) is negative in 2000 and is not significant with the simple spatial contiguity matrix ($W_{adj}$). However, in 2010, the coefficient becomes positive and significant at a 5% level with both weight matrices, indicating the increase in the proportion of older workers in the labor force, and they became a significant driving effect on productivity growth. As argued by Ang and Madsen [42], the productivity growth effects of elderly workers with higher education are substantially higher than those of their younger counterparts, due to the accumulation of experience and knowledge. Compared with their counterparts in 2000, the elderly workers born between 1951 and 1960 have a better chance to gain higher education. As the college entrance examination system in China was interrupted from 1966 to 1976 and was not restored until 1977, people born between 1941 to 1950 were often less exposed to higher education. As a result, the elderly workers in 2010 might have higher creativity and management quality to improve productivity than their counterparts in 2000. This change in the impact of different cohorts on productivity has also been observed in the United States [14].

Third, the coefficients of migration (MIG) become larger and significant in 2010, though they are positive in 2000, but not statistically significant (Table 5). This might suggest that migrants in 2000 are mainly unskilled surplus labor from rural areas (the typical "*Nongmin Gong*", or peasant workers) and they often work in labor intensive industries, such as clothing and construction, which contributes little to productivity improvement. In 2010, the proportion of skilled workers, such as high tech talents and college students, has increased substantially. Migration becomes an important way of human resource reallocation. Migration factor in 2010, hence, generated a significant impact on local productivity. In summary, human capital stock might be the most important population factor for productivity growth, and the impact of other factors largely depends on it.

Fourth, as for the other control variables, some meaningful findings emerge from the regression results as well. Fixed asset investment (INV) shows a significant negative effect in both years, which might be a result of China's economic policies changing during that time. For a long time after the reform and opening up, China served as a "world factory." China's economic growth has been mainly dependent on massive investments of labor, physical capital, mineral and energy resources. The development pattern with high material consumption is obviously harmful to productivity growth in 2000. In 2008, to ease the downward pressure on the economy caused by the world financial crisis, the Chinese government put forward the four trillion investment fiscal policy. The strong stimulus policy successfully completed the task of "maintaining the growth." Firm investment became less efficient due to the increase of bank loans and government subsidies [69]. The problems of a rigid economic structure, high energy consumption and high pollution in China have not been effectively addressed, and still play a restraining role in the productivity growth. The change in the symbol of government expenditure (GOV) and the change in the significance of industry structure (SEC or TER), might also be attributed to the change in economic policies during this period, for similar reasons. The openness (FDI) is always positive at the 99% confidence level, which is in line with previous studies [9,10]. In today's globalized world, openness promotes market expansion, technology introduction and talent exchange between countries, and it plays an increasingly important role in productivity growth.

Fifth, from the model summary in Table 6, compared with their corresponding OLS models, the log likelihood values become larger and AIC values become smaller in all spatial regression models, indicating the spatial models fit the data better. The direction and significance of the coefficients of most independent variables—including the same population variables, control variables and spatial lag term—in Model 2–4 are consistent with those in Model 1. This suggests that the findings are relatively robust and credible. However, despite the addition of the squared term, the population size is always insignificant concerning affecting productivity growth, which suggests that the inverted-U effect of population size on productivity growth does not exist at the prefecture level in China. On the other hand, the estimation results of population density and its squared term in Model 3 and Model 4 show that population density does affect productivity growth and that there is an inverted U-shaped relationship between them. In other words, when population density is below a threshold point, a higher density improves productivity growth. Once population density reaches and passes the threshold point, higher density exerts a negative impact on productivity growth. This result suggests that the concentration of the population, instead of the sheer number of population, is one of the primary demographic factors in driving the prefectures' productivity growth. The coefficient of human capital inequality has changed from significantly negative in 2000 to insignificantly positive in 2010. The Chinese government attaches great importance to education and has invested heavily to raise the national education level. In 2006, China fully implemented the nine year free compulsory education system to promote fairness and eliminate inequality in education. This is also reflected in the change in the human capital Gini coefficient, which has decreased from 0.250 in 2000 to 0.213 in 2010. Apparently, increased human capital equality practically removed the negative effect it had on TFP in the early 2000s, though the potential positive effect still needs to fully manifest in the future.

## 5. Conclusions

This study employs spatial data analysis methods, ESDA and the spatial autoregressive model, to investigate the change of global and local spatial autocorrelation of total factor productivity and examine the impact of demographic factors on productivity growth. By comparing the results of exploratory and empirical analysis in 2000 and 2010, we have the following main conclusions.

First, there is a strong spatial autocorrelation between total factor productivity among cities, and it has increased from 2000 to 2010. That means that the spatial spillover effect of productivity becomes stronger with the expansion of transportation networks and the improvement in other infrastructure. Local Moran's I significance maps show that total factor productivity has changed from a gradient pattern, decreasing from east to west in 2000, to a sandwichlike pattern in 2010.

Second, the role of population factors in productivity growth has changed from 2000 to 2010. According to the results of spatial regression models, population density (rather than population size) and human capital stock are important factors in both years and there is an inverted U-shaped relationship between population density and productivity growth. Due, largely, to the specific educational background of different cohorts, the growth effect of elderly workers and migration has changed from insignificant in 2000 to significantly positive in 2010. With the improvement in educational fairness in China, human capital inequality seems to be no longer a critical factor affecting productivity growth.

Third, among nonpopulation factors, fixed asset investment always has a significant negative effect, while openness is always the positive factor on productivity growth. In addition, the impact of industrial structure and government expenditure has changed from 2000 to 2010, which might be attributed to the special economic policies implemented during the study period.

These findings complement current literature on China's population and productivity and provide certain insights into demographic and economic policies. First, the spatial interdependence effect should be taken into account in regional policymaking. As shown in the results of spatial autocorrelation analysis, economic productivity growth in a city is closely related to the surrounding cities. Therefore, policies to promote productivity need to consider not only the local conditions, but also the opportunities and challenges of the surrounding regions. This means that there is a need for interregional cooperation and communication from a higher administrative level in order to narrow divergence among regions and promote common development. Second, population age structure, migration, and quality, rather than population size, are the core issues of present and future policies. The change in the impacts of population on productivity from 2000 to 2010 is the epitome of the periodic transition of the relationship between population and economy in China. According to the current trend, China's population size will decline in the next few years, while the proportion of the elderly, the proportion of urban residents and the proportion of the highly educated population will continue to rise. Relevant policies should be adapted to these demographic changes and make full use of the positive roles of demographic factors in promoting economic productivity and quality. For example, population regulation policies should be strictly implemented in megacities, such as Beijing, Shanghai, and Shenzhen, to avoid population density exceeding the threshold point. With the deepening of the economic reform and globalization, the Chinese government has been committed to the relaxation of the household registration system, promoting urbanization, and expanding investment in education and scientific research. It is believed that the migration and cluster of populations with high education levels and skills will improve China's economic productivity and quality in the future.

This study has several limitations that are expected to be improved in the future work. First, as aforementioned, the census data we used in the study needs to be updated when the newest data is released to provide a more up to date picture of the population-productivity dynamics. Additionally, although the spatial lag regression model has more advantages than the traditional OLS model in analyzing spatial data, it is still a global

model in essence. This means that the impact of each population factor on productivity growth is spatially unchanged, so the coefficients are constant. However, due to regional differences and spatial diversity, the real economic behavior in China is geographically changeable and there is no stable spatial pattern of total factor productivity [52]. Our future research will attempt to evaluate and model such potential spatially varying relationships to gain detailed insights into China's demographic and economic development.

**Author Contributions:** Conceptualization, X.W. (Xiaoxi Wang), Y.Z. and D.Y.; methodology, X.W. (Xiaoxi Wang), Y.Z. and D.Y.; software, X.W. (Xiaoxi Wang) and D.Y.; validation, X.W. (Xiaoxi Wang), Y.Z., D.Y., X.W. (Xiwei Wu) and D.L.; formal analysis, X.W. (Xiaoxi Wang) and D.Y.; Investigation: X.W. (Xiaoxi Wang) and Y.Z.; resources, X.W. (Xiaoxi Wang), Y.Z., X.W. (Xiwei Wu) and D.L.; data curation, X.W. (Xiaoxi Wang); writing—original draft preparation: X.W. (Xiaoxi Wang); writing—review and editing: X.W. (Xiaoxi Wang), Y.Z. and D.Y.; visualization: X.W. (Xiaoxi Wang); supervision, Y.Z. and D.Y.; project administration: Y.Z. and D.Y.; funding acquisition: Y.Z. and D.Y. All authors have read and agreed to the published version of the manuscript.

**Funding:** This research was funded by National Natural Science Foundation of China grant numbers 71373275 and 41461035.

**Institutional Review Board Statement:** Not applicable.

**Informed Consent Statement:** Not applicable.

**Data Availability Statement:** Not applicable.

**Conflicts of Interest:** The authors declare no conflict of interest.

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
