# Peer review of "Changes in Demographic Factors’ Influence on Regional Productivity Growth: Empirical Evidence from China, 2000–2010"

_sustainability, doi:10.3390/su14074195_

Round 1

Reviewer 1 Report

General comment:

The paper addresses a very interesting research topic, that is, the effects of demographic factors on regional productivity growth. Nevertheless, the theoretical framework needs to be reinforced, and the research hypotheses need to be directly derived from previous theoretical and empirical frameworks. Afterwards, the discussion should be revisited by contrasting the findings with the research hypotheses raised from the referred framework.

Specific comments:

In order to improve the global quality of the manuscript, the following specific suggestions and recommendations are also made available:

(1) The motivation in the introductory item needs further improvement and refinement, by clearly stating the need for accomplishing this research and taking into consideration the scientific state-of-the-art.

(2) The way how references are used along the manuscript needs to be uniformized.

(3) Table 1 is not required.

(4) It is recommended the inclusion of a robustness check, using for example control variables, or if it is possible, a set of instrumental variables. 

(5) The limitations of the methodology and empirical approach need to be displayed.

(6) The policy implications derived from the current approach need to be included in the concluding remarks, as well extending the suggestions of future research.

Author Response

Reviewer 1: The paper addresses a very interesting research topic, that is, the effects of demographic factors on regional productivity growth. Nevertheless, the theoretical framework needs to be reinforced, and the research hypotheses need to be directly derived from previous theoretical and empirical frameworks. Afterwards, the discussion should be revisited by contrasting the findings with the research hypotheses raised from the referred framework.

Reply: Thanks for your suggestion. In this revision, we have further reviewed and summarized previous theoretical and empirical studies and found that counteracting forces of demographic factors exist at the same time. Additionally, we argued that the impacts of demographic factors on productivity growth varies at different times. As suggested, we derived the the research hypotheses prior to the discussion. After data analysis and empirical regression, the results are interpreted and discussed in combination with the actual development situation so as to get meaningful findings and conclusions, which verifies the hypothesis.

Specific comments:

In order to improve the global quality of the manuscript, the following specific suggestions and recommendations are also made available:

(1) The motivation in the introductory item needs further improvement and refinement, by clearly stating the need for accomplishing this research and taking into consideration the scientific state-of-the-art.

Reply: Thank you for your advice. In this round of revision, in the Introduction Section, we state the motivation of this study from three aspects. First, the importance of improving total factor productivity for China's economic development is emphasized. Chinese government pays more and more attention to promoting total factor productivity in plannings and documents. Second, we briefly show the changes in China's population and the possible impact on economic productivity. Moreover, there are some limitations in previous studies on China’s population and productivity. Third, the contributions and values of this study are introduced in detail. We think our work that investigating the impact of demographic factors on economic productivity from the perspective of cross-temporal comparison will help to figure out the practical confusions and put forward proper policy suggestions.

(2) The way how references are used along the manuscript needs to be uniformized.

Reply: We are sorry for the mistake of reference style caused by our carelessness. In the revised version, we have strictly followed the style guides of sustainability journal to modify our citations and bibliographies. Please see the revised manuscript for details.

(3) Table 1 is not required.

Reply: Thanks for your suggestion. The intention of setting Table 1 is to summarize the effects of different demographic factors to make the theoretical analysis clearer. Considering that the textual description of the relevant theories is clear enough, so in the updated manuscript, we decided to delete Table 1 to avoid redundancy.

(4) It is recommended the inclusion of a robustness check, using for example control variables, or if it is possible, a set of instrumental variables.

Reply: Thanks for your advice. According to the existing studies, governmental investment (represented by fixed assets investment), industrial structures (portion of secondary and tertiary industries in the economy), government financial expenses, and the overall openness are selected as control variables in empirical models. The regression results of these control variables are also discussed in detail. In addition, the application of two different spatial weight matrices (simple binary matrix and inverse distance weight matrix) can be seen as a simple robustness check. When the spatial relationship between different cities changes, the regression results show robustness and do not change subversively. As for instrumental variables, data availability is the primary reason that we decided not to employ instrumental variable analysis. We hope you may understand our difficulty and we stated such in this revision.

(5) The limitations of the methodology and empirical approach need to be displayed.

Reply: Thanks for your advice. In the revised version, we have displayed the limitation of empirical approach at the end of the article. Specifically, spatial lag regression model we used in the study is a global model in essence, though it is more suitable for analyzing spatial data than traditional OLS model. The coefficients of global model are constant because the effects of explanatory variables are considered to be spatially unchanged. As a result, the spatial heterogeneity of influence mechanism is masked. Therefore, in follow-up work, we will employ geographical weighting regression to model potential spatially varying relationship. It is expected to provide more interesting and meaningful findings about China’s demographic factors and economic productivity.

(6) The policy implications derived from the current approach need to be included in the concluding remarks, as well extending the suggestions of future research.

Reply: Thank you very much for your suggestion. We have noticed the importance of policy suggestions for the practical value of this study. Relevant contents have been added in the revised manuscript. According to the results and findings, we proposed two policy implications. First, spatial interdependence effect should be taken into account in regional policy-making. Second, demographic, and economic policies should be adapted to the population changes and make full use of the positive roles of demographic factors in promoting economic productivity and quality. More details can be seen in the Section 5.

Reviewer 2 Report

The authors of the paper "The change of demographic factors' influences on regional productivity growth: empirical evidence from China, 2000 - 2010" present a relevant topic, namely "analysis of changes in global and local spatial dependence of total factor productivity", a particularly important aspect for the global economy with the direct involvement of the population density and the increase of productivity, which makes the work have the effect of multiplication.

The bibliographic sources, citations, concepts, theories and hypotheses established in the paper are appropriately used by the authors of the research. Specifically, those related to the effect of population size on productivity growth (24, 29). At the same time, the authors highlight works that support "the theory of the dualistic economic structure proposed by Arthur Lewis (34) which is often used to explain the phenomenon of migration on productivity."

The research methodology is simplistically presented, respectively the data used by the authors in the analysis of this study are those from the 2000 and 2010 censuses, considered to be the most recent. However, given that we are in 2022, as well as the fact that from 2010 to 2022 there have been demographic changes that influence the data of the authors' analysis and could be a limitation of the study. Variables and their control models, influencing factor of total factor productivity, exploratory spatial data analysis (ESDA), Moran index, spatial regression models, spatial specification lag (SLM), spatial error (SEM) and model autoregressive with spatial lag.

The results of the research are presented by the authors of the research based on the interpretation of the data and show that "there is a significant spatial autocorrelation in OLS regression residues" and that "population size does not seem to have any effect on productivity growth in both years." and "migration coefficients (MIGs) become higher and more significant in 2010, although positive in 2000 but not statistically significant," and "spatial regression models, indicating that spatial models fit data better." However, we suggest that the authors of the research highlight the main scientific results as a personal contribution to the scientific literature, given that the results are adequately presented, but compared to the 2010 data (which would be a limitation) and we appreciate the scientific side of the work.

The conclusions are presented by the authors of the research, respectively the authors emphasize that "they investigate the change of global and local spatial autocorrelation of total factor productivity and to examine the impact of population factors on productivity growth." Moreover, the authors of the study also present the involvement in future research, as well as the aspects from the applicative point of view. However, we suggest that the authors present the limitations of the study. Moreover, as we mentioned in the results chapter, we appreciate that the personal scientific results that contribute to the scientific literature should be highlighted.

We congratulate the research team, we suggest the revision of the paper according to the above mentioned.

Author Response

Reviewer 2: The authors of the paper "The change of demographic factors' influences on regional productivity growth: empirical evidence from China, 2000 - 2010" present a relevant topic, namely "analysis of changes in global and local spatial dependence of total factor productivity", a particularly important aspect for the global economy with the direct involvement of the population density and the increase of productivity, which makes the work have the effect of multiplication.

The bibliographic sources, citations, concepts, theories and hypotheses established in the paper are appropriately used by the authors of the research. Specifically, those related to the effect of population size on productivity growth (24, 29). At the same time, the authors highlight works that support "the theory of the dualistic economic structure proposed by Arthur Lewis (34) which is often used to explain the phenomenon of migration on productivity."

Reply: Thank you for your kind comments. We strive to make the best of our data and analysis and hopefully this study will contribute significantly to the scholarly community.

The research methodology is simplistically presented, respectively the data used by the authors in the analysis of this study are those from the 2000 and 2010 censuses, considered to be the most recent. However, given that we are in 2022, as well as the fact that from 2010 to 2022 there have been demographic changes that influence the data of the authors' analysis and could be a limitation of the study. Variables and their control models, influencing factor of total factor productivity, exploratory spatial data analysis (ESDA), Moran index, spatial regression models, spatial specification lag (SLM), spatial error (SEM) and model autoregressive with spatial lag.

The results of the research are presented by the authors of the research based on the interpretation of the data and show that "there is a significant spatial autocorrelation in OLS regression residues" and that "population size does not seem to have any effect on productivity growth in both years." and "migration coefficients (MIGs) become higher and more significant in 2010, although positive in 2000 but not statistically significant," and "spatial regression models, indicating that spatial models fit data better." However, we suggest that the authors of the research highlight the main scientific results as a personal contribution to the scientific literature, given that the results are adequately presented, but compared to the 2010 data (which would be a limitation) and we appreciate the scientific side of the work.

The conclusions are presented by the authors of the research, respectively the authors emphasize that "they investigate the change of global and local spatial autocorrelation of total factor productivity and to examine the impact of population factors on productivity growth." Moreover, the authors of the study also present the involvement in future research, as well as the aspects from the applicative point of view. However, we suggest that the authors present the limitations of the study. Moreover, as we mentioned in the results chapter, we appreciate that the personal scientific results that contribute to the scientific literature should be highlighted.

We congratulate the research team, we suggest the revision of the paper according to the above mentioned.

Reply: Thank you for your detailed comments. This is very helpful. In this round of revision, we have clearly stated the limitation of our relatively outdated data. As you mentioned, the data we used in the study are those from the 2000 and 2010 censuses. There are two reasons why we did that. First, detailed demographic data are indispensable to calculate some variables in empirical models, such as the number of net immigrants and human capital Gini coefficient. However, it is difficult to obtain some data, such as population by educational level and migration status, in non-census years. Second, although 2020 census has been conducted in China, specific data at the city-level have not yet been released when the study was conducted. Therefore, the data in 2000 and 2010 are regarded as the latest available data in our study. It is true that population and economic productivity in China have experienced great changes within a time span from 2010 to today, the study could still shed light on how demographic factors impact productivity in China. In the revised version, we have emphasized the data limitation at the end of the article.

At present, China's population and economy are undergoing great changes. Population age structure, migration, and quality rather than population size is becoming core issues in policymaking. Improving total factor productivity has become an important channel to jump out of the middle-income trap and promote economy quality. The transition of the relationship between population and economy aroused our interest to carry out this study. This study employs city-level data and spatial data analysis methods and models to investigate the change of demographic factors’ impacts on regional productivity growth. It is hoped that our work could provide more thinking and enlightenment for future studies.

Additionally, another limitation of this study is about empirical approach. As we displayed in the revised manuscript, although spatial lag regression model has advantages over traditional OLS model in analyzing spatial data, it is still a global model in essence. This means that the impact of explanatory variables on dependent variable is spatially unchanged, so the coefficients are constant. Spatial heterogeneity is another important characteristic of spatial data. We intend to employ geographical weighting regression to model potential spatially varying relationship between demographic factors and economic productivity in follow-up work.

In this round of revision, we have emphasized the contributions of this study that are primarily reflected in two aspects. First, nation-level or province-level analyses are common in previous studies and there is almost no study conducted at city-level. Moreover, the change of population is comprehensive, but researchers usually explore only one demographic factor instead of a whole array of demographic factors. Our study uses data from 358 cities and analyzes the impacts of different demographic factors on economic productivity. The findings could enrich existing literature from theoretical and empirical perspectives. Second, in our revised manuscript, we proposed the policy implications of this study based on analysis results and findings. Spatial interdependence effect should be taken into account in regional policymaking. Relevant policies should be adopted to the population changes and make full use of the positive roles of demographic factors in promoting economic productivity and quality.

We hope our revisions can meet your expectations and thanks again for your efforts in reviewing.

Round 2

Reviewer 1 Report

Considering the changes made, it is recommended the acceptance of the manuscript.